# XBT, ARGO Float and Ship-Based CTD Profiles Intercompared under Strict Space-Time Conditions in the Mediterranean Sea: Assessment of Metrological Comparability

**Andrea Bordone [1], Francesca Pennecchi [2] , Giancarlo Raiteri [1,*], Luca Repetti [3] and Franco Reseghetti [1]**

[1] ENEA—Marine Environment Research Centre S. Teresa, 19032 Pozzuolo di Lerici (SP), Italy; andrea.bordone@enea.it (A.B.); franco.reseghetti@enea.it (F.R.)

[2] INRiM—Istituto Nazionale di Ricerca Metrologica, Strada delle Cacce 91, 10135 Torino, Italy; f.pennecchi@inrim.it

[3] IIM—Istituto Idrografico della Marina Militare, Passo dell'Osservatorio 4, 16134 Genova, Italy; luca_repetti@marina.difesa.it

\* Correspondence: giancarlo.raiteri@enea.it; Tel.: +39-018-797-8271

**Abstract:** Accurate measurement of temperature and salinity is a fundamental task with heavy implications in all the possible applications of the currently available datasets, for example, in the study of climate changes and modeling of ocean dynamics. In this work, the reliability of measurements obtained by oceanographic devices (eXpendable BathyThermographs, Argo floats and Conductivity-Temperature-Depth sensors) is analyzed by means of an intercomparison exercise. As a first step, temperature profiles from XBT probes, deployed by commercial ships crossing the Ligurian and Tyrrhenian seas during the Ship of Opportunity Program (SOOP), were matched with profiles from Argo floats quasi-collocated in space and time. Attention was then paid to temperature/salinity profiling Argo floats. Since Argo floats usually are not recovered and should last up to five years without any re-calibration, their onboard sensors may suffer some drift and/or offset. In the literature, refined methods were developed to post-process Argo data, in order to correct the response of their profiling CTD sensors, in particular adjusting the salinity drift. The core of this delayed-mode quality control is the comparison of Argo data with reference climatology. At the same time, the experimental comparison of Argo profiles with ship-based CTD profiles, matched in space and time, is still of great importance. Therefore, an overall comparison of Argo floats vs. shipboard CTDs was performed, in terms of temperature and salinity profiles in the whole Mediterranean Sea, under space-time matching conditions as strict as possible. Performed analyses provided interesting results. XBT profiles confirmed that below 100 m depth the accordance with Argo data is reasonably good, with a small positive bias (close to 0.05 °C) and a standard deviation equal to about 0.10 °C. Similarly, side-by-side comparisons vs. CTD profiles confirmed the good quality of Argo measurements; the evidence of a drift in time was found, but at a level of about E−05 unit/day, so being reasonably negligible on the Argo time-scale. XBT, Argo and CTD users are therefore encouraged to take into account these results as a good indicator of the uncertainties associated with such devices in the Mediterranean Sea, for the analyzed period, in all the climatological applications.

**Keywords:** XBT; ARGO float and ship-based CTD intercomparison; temperature and salinity profiles; space-time matching conditions; metrological comparability

## 1. Introduction

Temperature and salinity are very important quantities to study the properties of seawater and its changes over time. Different devices are available, based on different recording techniques and physical effects: their accuracy is quite different so that when the measures of the same quantity from different sensors show a spread of values, it can be difficult to use such values correctly. In this paper, the performances of popular sensors measuring temperature and salinity, namely eXpendable BathyThermographs (XBTs), Argo floats and ship-based Conductivity-Temperature-Depth sensors (CTDs), were intercompared. The main purpose was to assess the metrological comparability of such transducers (in about the last two decades) in the Mediterranean Sea, which is a marginal sea with both unusually high temperature and salinity values and a peculiar shape of these profiles.

An XBT system, including an expendable probe falling in water, a launcher with a connecting cable and a data acquisition unit, is a well-known instrument to measure temperature (*t*) profiles in oceanography [1–4]. XBT can be considered as a cheap, versatile, and easy to use transducer. Due to these advantages, between 1970 and 1990 with around 90 thousand probes per year, XBTs measured most of the temperature data in the upper 2000 m of the oceans, in particular, along the main commercial ship lines (Figure 1). Consequently, there is considerable literature on the subject [5,6].

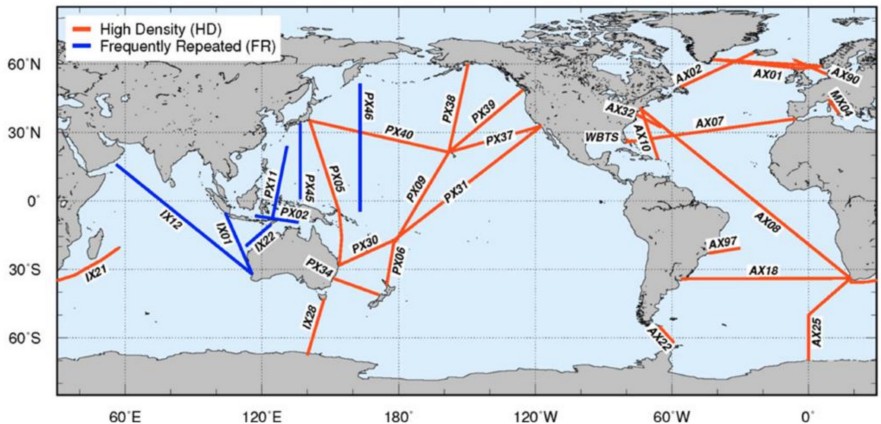

**Figure 1.** Location of XBT transects, as defined by the XBT Science Team [6].

Nowadays, the quantity of XBT probes launched annually has been considerably reduced (about 15 thousand per year) due to the widespread network of temperature/salinity profiling floats, known as Argo, that has become a fundamental component of the ocean observing system [7–10]. Nevertheless, XBT transects are still considered a useful, complementary source of oceanographic information: they provided up to now, in fact, very long records of temperature observations across ocean basins, that are of crucial importance in research related to ocean heat content and current variability, together with water mass and heat transport [6]. In this context, climatologists highlighted both the importance of historical XBT datasets and the need to accurately evaluate their measurement uncertainties, which are fundamental for climatological analyses [11]. For this purpose, a detailed comparison of XBT vs. Argo temperature profiles was considered critical, according to specific literature related to XBT data quality improvement [12–14]. Comparison between quasi-collocated and quasi-simultaneous XBT and Argo measurements was then focused on both Tyrrhenian and northeastern Ligurian seas, along the MX04 XBT transect (Genoa to Palermo, mapped in Figure 1), historically managed by ENEA S. Teresa Research Centre (since September 1999) in the context of Ship of Opportunity Program (SOOP) [6,15]. To date, about 90 transects have been completed, resulting in over 3000 profiles. In terms of Argo profiling floats (Argo in the following), they mainly measure temperature and salinity ($S_P$, practical salinity) of world oceans from an array of more than 3000 underwater robots. They drift and are carried by currents at a selected parking depth. Then, usually at intervals of 5, 8 or 10 days, they go down to a greater depth before rising to the surface. During the ascent, *t* and $S_P$ values are recorded:

at the surface, pairs at selected values are transmitted back to the Thematic Assembly Centers via satellites. Finally, they return to parking depth to start a new measuring cycle [7]. Argo floats host CTD sensors (usually model SBE 41/41CP) calibrated on bench before deployment, whose nominal accuracies are actually comparable to accuracies of shipboard CTDs [7]. Temperature measures in the Argo CTD profiles are declared to be accurate to ±0.002 °C, while pressure ones are accurate to ±2.4 dbar [16]. For salinity measures, it has to be considered that the conductivity cell is more sensitive to possible drift and/or offset (due, for example, to fouling that accumulates over the years, varying the dimension of the cell itself). Therefore, $S_P$ data delivered in real time are declared to be accurate to ±0.01 PSU [16]. However, in a second stage, salinity measures are usually post-processed and corrected by expert examination, comparing Argo data vs. historical data used to estimate the background climatological salinity (mainly acquired by older Argo floats or ship-based CTD data). Salinity data are in this way post-validated (or adjusted) following a method known as Delayed Mode Quality Control [17–19]. This refined method is tightly connected to high quality ship-based CTD measures, to which Argo profile should be always compared in order to maintain, in a reasonable way, the necessary metrological traceability [20–22]. The aim of the present work was also to show the main results obtained from comparing (adjusted and not-adjusted) Argo profiles vs. ship-based CTD profiles, mated under strict space and time matching conditions, not so common in the literature on such a large scale [23]. The comparison was performed starting from 2000 (the year in which Argo deployments began) and taking into account the whole Mediterranean Sea, where the overall coordination of profiling float operations is in charge of MedArgo program (together with Argo data control and distribution) [24–27].

## 2. Materials and Methods

Analysis performed in this work on large datasets, deposited in publicly available databases, is described in detail in Section 2.1 for XBT vs. Argo and in Section 2.2 for Argo vs. ship-based CTD comparisons, respectively.

### 2.1. XBT vs. Argo Pairing

XBT and Argo profiles were downloaded from dedicated online databases ([28,29] and [7], respectively). Each XBT profile underwent a quality control process in which a series of tests assessed the quality of the measurements (i.e., presence of spikes, constant value profiles, extreme depth (*d*) and temperature values, improper dates and locations, vertical gradients and inversions, wire breaks, seafloor contact, etc. [6]). In particular, for the XBT data starting from the cruise of the 29[th] of July 2010, a check of performances of the acquisition system was available through the calibration of the system itself by a test canister working at two reference temperatures. This control procedure was performed immediately before the XBT launches started and immediately after the launch of the last XBT probe. A slight deviation from the reference temperature values was often verified, especially at high temperatures, sometimes even combined with a different result between the values read at the beginning and end of the XBT deployment, a symptom of a possible temporal drift of the system. It was therefore decided to correct the XBT profile data by applying an algorithm that linearly corrects the XBT temperature reading according to the time elapsed since the first launch (to take into account the time drift) combined with a further linear correction as a function of the deviation from the reference temperature values. A detailed analysis of all this correction is in preparation [30]. Following the indications for the first Rossby radius of deformation indicated in [31], XBT and Argo profiles were matched in pairs, in which the former was considered as the reference in space and time to which the latter was compared (i.e., position and instant of the XBT deployment were considered as the position and time zero, respectively). The matching 3D-space and time conditions were chosen as follows (it has to be underlined that two different time windows were considered, in order to have a first dataset comparable to previous study [12] and a second dataset with a more restrictive matching condition in time):

- ΔLatitude: ±0.10°;

- ΔLongitude: ±0.15°;
- Δtime: ±7 days and ±1 day (nominal intervals);
- Δdepth: ±1 m.

The coordinate differences allowed to have a maximum distance of about 12 km between each XBT profile and the matched Argo one (the Rossby radius in the area of interest is about 10 km). The considered period spans from the 16th of August 2004 up to the 19th of March 2019. As a function of the two different time windows, the dataset could be divided as follows.

2.1.1. XBT vs. Argo Pairing: Dataset Obtained in the Large Time Window (±7 days)

In total, 147 XBT vs. Argo paired profiles were found, satisfying the imposed matching conditions (mean spatial matching equal to (9.4 ± 3.1) km). In Figures 2 and 3, the spatial distribution of XBT-Argo considered pairs and an example of matched temperature profiles are reported, respectively. The actual number of different XBT probes was equal to 94, mainly Deep Blue type (DB). More in detail, there were thirteen T4, seven T5, one T6, two T10 and seventy-one DB types [32]: this distribution reflected in some way the amount of different deployed XBT probes. The actual number of different Argo floats involved in the comparison was equal to 24, originating 127 profiles.

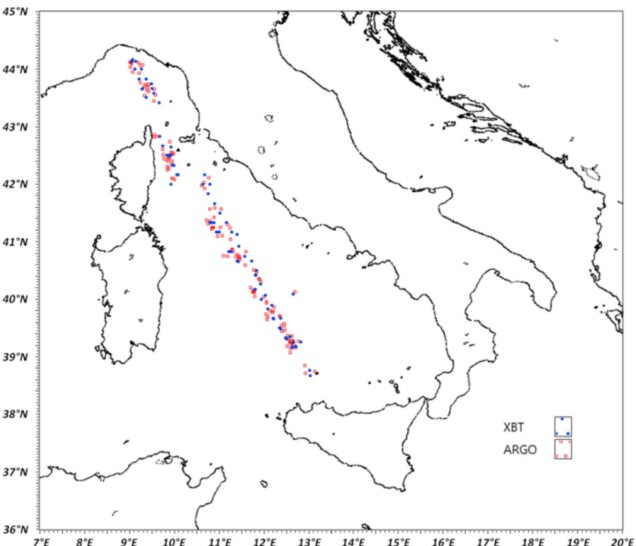

**Figure 2.** Spatial distribution of the XBT-Argo pairs analyzed along the MX04 transect in about 15 years.

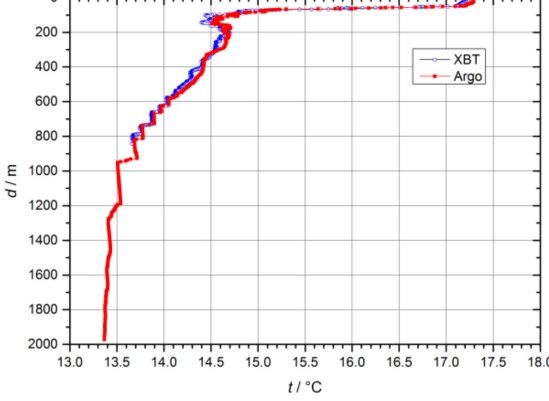

**Figure 3.** Example of quasi-collocated and quasi simultaneous XBT and Argo profiles. XBT (Deep Blue)—Date: 2018/12/11, Time: 14:05:50, Lon: 12.6322°E, Lat: 39.1667°N. Argo (#6902903)—Date: 2018/12/14, Time: 10:19:00, Lon: 12.6344°E, Lat: 39.2362°N.

As detailed in the following, full profiles and, separately the 0–100 m and *d* > 100 m regions of the sea water column, were considered. This is due to the well-known depth error that affects the XBT measurements (estimated as the greater value between 5 m and 2% of the depth itself [32]), usually well evident at the start of the upper seasonal thermocline. The same analyses were also repeated on different data subsets depending on the XBT types. An overall number of 15,740 matched temperature values were found, quasi-collocated in depth along the water column (within ±1 m). To give evidence of the numerosity of the analyzed sample, in Figure 4 the number of XBT-Argo pairs is reported, divided per XBT type and depth interval.

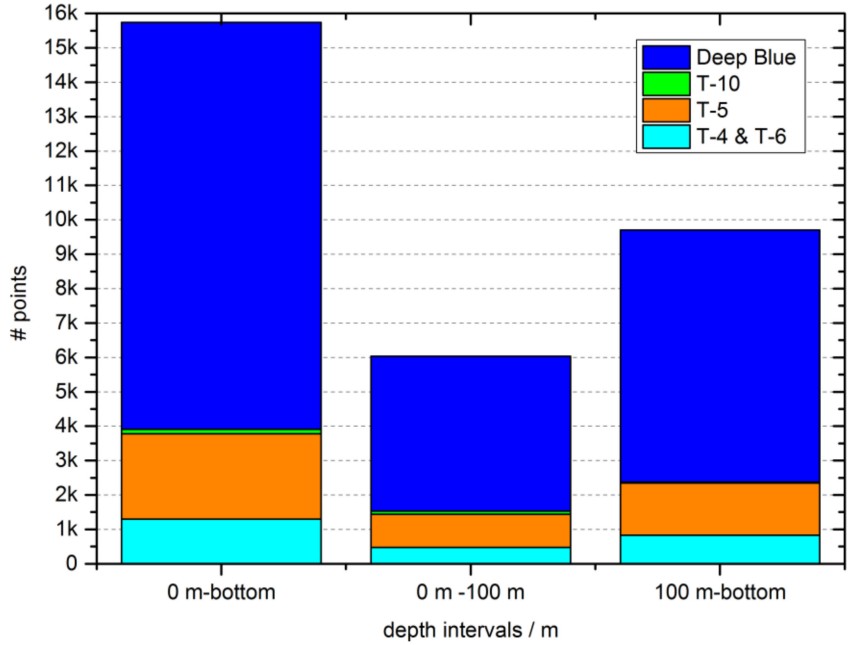

**Figure 4.** Number of matches XBT vs. Argo divided per XBT type and depth intervals (time window ±7 days).

### 2.1.2. XBT vs. Argo Pairing: Dataset Obtained in the Strict Time Window (±1 day)

By applying the same space matching conditions, but with a stricter time window, 31 XBT vs. Argo paired profiles were found (mean spatial matching equal to (9.3 ± 3.4) km). The actual number of different XBT probes was equal to 31 (six T4, two T5 and twenty-three DB types), while 10 different Argo floats were involved in the comparison, originating 24 profiles. An overall number of 2601 matched temperature values were found (quasi-collocated in depth along the water column, within ±1 m). This is a subset of the previously presented dataset.

### 2.2. Argo vs. Ship-Based CTD Pairing

Both Argo and CTD profiles were downloaded from WOD (World Ocean Database, release WOD18-March 2019 [29]), according to the following searching criteria:

- Year: from 2000 to 2018 (all months and days);
- Longitude range: from 6°W to 36°E;
- Latitude range: from 30°N to 44.5°N;
- Measured variables: $t$, $S_P$;
- Dataset: CTD, PFL (Argo Profiling Floats).

As a result, 60,838 total casts were obtained by WOD, divided as follows:

- no. 5664 CTD casts;
- no. 55,174 Argo casts.

Each cast was then checked in order to filter data according to both the available quality flags, i.e., those supplied by WOD itself and those by the data originator, respectively [33,34]. As a consequence, for each CTD cast, only data flagged by WOD with flag "0" were selected: this check was performed for the entire cast (where "0" means "accepted cast") and for the individual observations of depth, temperature and salinity (where "0" means "accepted value"). At the same time, applying the method of a logical AND, Argo casts flagged by the originator with flag "1" (that means "good data") and by WOD with flag "0", were selected for the comparison.

After collecting these two datasets, following again the indications for the first Rossby radius of deformation indicated in [31], Argo and CTD profiles were matched under the following 3D space-time conditions:

- ΔLatitude: ±0.10°;
- ΔLongitude: ±0.15°;
- Δdepth (for each $t$ and $S_P$ values in the matched profiles): ±1 m;
- Δtime: ±1 day.

By adding the requirement that each couple of Argo and CTD profiles satisfying the previous conditions had in common at least ten values matched along the entire profile, the dataset was at last restricted to:

- Longitude range: from 3.097°E to 32.720°E;
- Latitude range: from 33.563°N to 43.533°N;
- Time period: from the 2nd of April 2006 to the 6th of June 2018;
- no. of profiles matched (Argo vs. CTD): 360;
- no. of Argo individual profiles: 96;
- no. of (ship-based) CTD individual profiles: 135 (for each profile, the CTD type is declared as "unknown" in the WOD database);
- no. of Argo floats involved: 47 (whose subdivision into models is reported in Table 1).

In the box plot shown in Figure 5, the actual distances in space and time between Argo and CTD profiles are reported; it can be noted that about 75% of matched profiles were separated by less than 12 km in space (mean 7.5 km) and about 24 h in time (mean 16.6 h).

Taking into account the relatively high number of matching profiles, these can be considered as reasonably strict space-time matching conditions, if compared with those reported in the literature: e.g., five Argo-CTD matched profile were considered in [35], where distance varies from 3.0 km to about 17.4 km (mean 8.2 km) and separation in time spans from 26.9 h to 41.9 h (mean 35 h). In other works, the space-time limits are taken as 100 km and 10 days [36] (for 38 matched profiles) or six days [37] (but for more than 500 matchups).

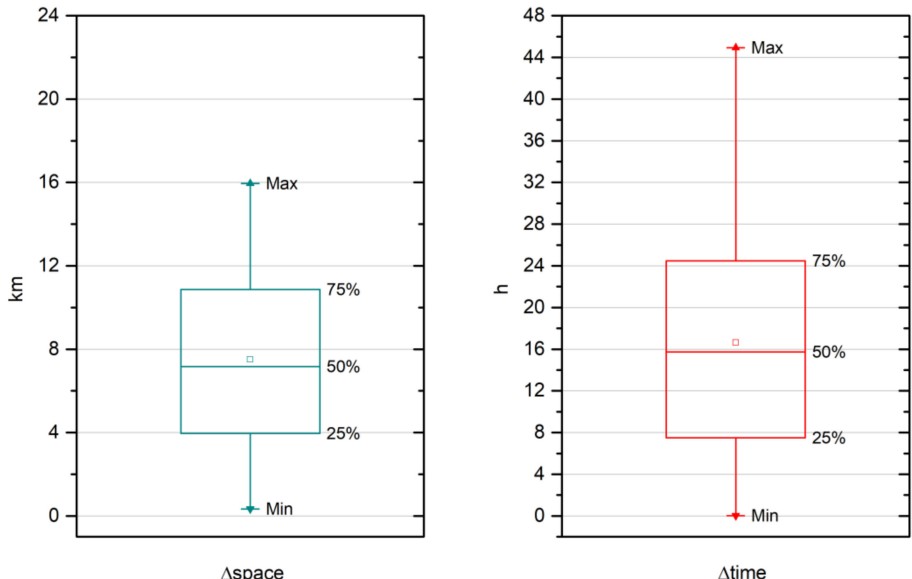

**Figure 5.** Distances in space (**left**) and time (**right**) between collected Argo and CTD profiles (mean values are indicated by empty squares).

In the map in Figure 6, the distribution in space and time of involved Argo floats is reported.

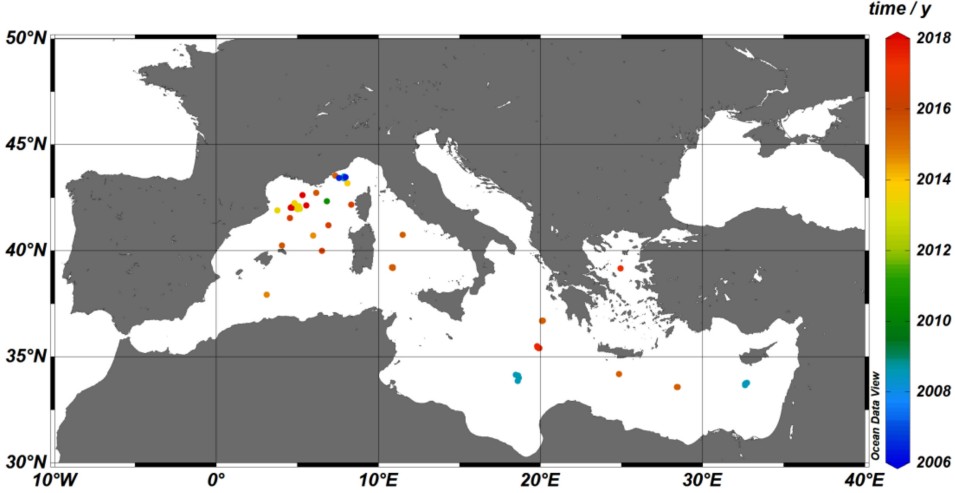

**Figure 6.** Positions of considered Argo floats, per year, in the Mediterranean Sea.

In Table 1, Argo models involved in this comparison are listed and counted (for more details, see [38]).

**Table 1.** Argo different models considered in the comparison vs. ship-based CTDs.

| Argo Model | # PFL | Argo Model | # PFL |
|---|---|---|---|
| APEX | 1 | PROVOR | 7 |
| ARVOR | 5 | PROVOR CTS2 | 2 |
| ARVOR A3 | 1 | PROVOR CTS31-DO | 2 |
| ARVOR DO | 2 | PROVOR CTS3-DO | 3 |
| ARVOR-I | 2 | PROVOR-II | 1 |
| ARVOR-N | 3 | PROVOR-III | 18 |



## 3. Results

In analogy with the previous sections, results of the comparison are presented according to the two different types of comparison performed.

### 3.1. XBT vs. Argo Comparison

3.1.1. XBT vs. Argo Comparison: Results Obtained in the Large Time Window (±7 Days)

In Figure 7 all the temperature differences (XBT-Argo) are reported as a function of depth, while in Figure 8 an overall summary of the corresponding results is shown (where *n* is the sample size, i.e., the number of considered pairs).

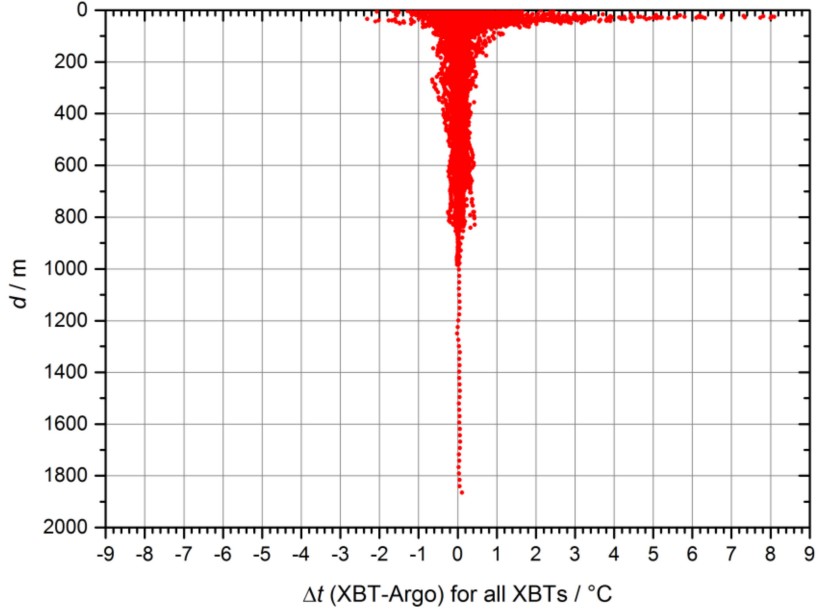

**Figure 7.** All temperature differences (XBT-Argo) vs. depth. Note the large values close to the surface.

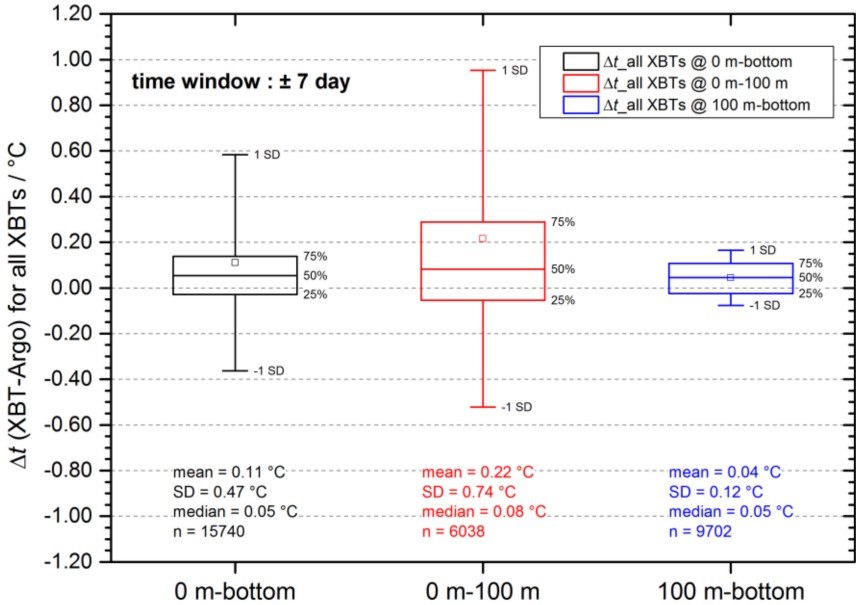

**Figure 8.** Box plot of temperature differences (XBT-Argo). Mean values are indicated by empty squares.

The mean $\Delta t$ calculated over the whole water column was +0.11 °C (but the median, being less sensitive to outliers, was +0.05 °C), with a standard deviation (SD) equal to 0.47 °C.

If the surface layer region 0–100 m (the typical thermocline region) was excluded, then mean $\Delta t$ would become equal to +0.04 °C (in practice the same value as for the median), with a SD value of 0.12 °C.

Detailed results, differentiated by XBT type, are shown in Tables 2–4 for depths 0 m-bottom, 0–100 m and 100 m-bottom, respectively.

**Table 2.** $\Delta t$ *(XBT-Argo) for each XBT type involved. Depth: 0 m-bottom (±7 days).*

|  | T-4 & T-6 | T5 | T10 | Deep Blue |
|---|---|---|---|---|
| # matched points | 1301 | 2481 | 131 | 11,827 |
| Max $\Delta t$ (°C) | 4.29 | 8.10 | 1.20 | 8.01 |
| Min $\Delta t$ (°C) | −1.09 | −1.65 | −0.60 | −2.31 |
| Mean $\Delta t$ (°C) | 0.11 | 0.13 | 0.33 | 0.10 |
| SD (°C) | 0.36 | 0.59 | 0.31 | 0.46 |
| Median $\Delta t$ (°C) | 0.07 | 0.02 | 0.25 | 0.06 |

**Table 3.** $\Delta t$ *(XBT-Argo) for each XBT type involved. Depth: 0–100 m (±7 days).*

|  | T-4 & T-6 | T5 | T10 | Deep Blue |
|---|---|---|---|---|
| # matched points | 475 | 961 | 100 | 4502 |
| Max $\Delta t$ (°C) | 4.29 | 8.10 | 1.20 | 8.01 |
| Min $\Delta t$ (°C) | −1.09 | −1.65 | −0.60 | −2.31 |
| Mean $\Delta t$ (°C) | 0.21 | 0.31 | 0.36 | 0.19 |
| SD (°C) | 0.57 | 0.91 | 0.35 | 0.72 |
| Median $\Delta t$ (°C) | 0.01 | 0.11 | 0.45 | 0.07 |

**Table 4.** $\Delta t$ *(XBT-Argo) for each XBT type involved. Depth: 100 m-bottom (±7 days).*

|  | T-4 & T-6 | T5 | T10 | Deep Blue |
|---|---|---|---|---|
| # matched points | 826 | 1520 | 31 | 7325 |
| Max $\Delta t$ (°C) | 0.48 | 0.57 | 0.42 | 0.93 |
| Min $\Delta t$ (°C) | −0.62 | −0.26 | 0.09 | −0.65 |
| Mean $\Delta t$ (°C) | 0.05 | 0.01 | 0.23 | 0.05 |
| SD (°C) | 0.12 | 0.12 | 0.08 | 0.12 |
| Median $\Delta t$ (°C) | 0.06 | −0.01 | 0.22 | 0.05 |

By considering Table 4, T5-type XBTs showed the best results in accuracy below 100 m (where also the dispersion value is reduced, due to a smaller temperature variability with depth along the water column).

### 3.1.2. XBT vs. Argo Comparison: Results Obtained in the Strict Time Window (±1 Day)

In Figure 9, a summary of the obtained results is shown.

By comparing values reported in Figures 8 and 9, it can be noted that a more strict matching condition on time has no significant influence on mean or median of temperature differences (for each depth interval considered); on the contrary, a slight reduction of SD values is evident (about 10%), showing an improved agreement between XBT and Argo measurements which are matched

in a stricter time window. Detailed results, differentiated by XBT type, are shown in Tables 5–7 for depths 0 m-bottom, 0–100 m and 100 m-bottom, respectively (no T10 type found in this case). Here again, T5-type XBTs showed the best results in accuracy below 100 m. In order to give evidence to the best behavior of T5 type, if compared with other XBT types, values of differences of the matched temperature values were plotted vs. depth (*d* > 100 m) in Figure 10.

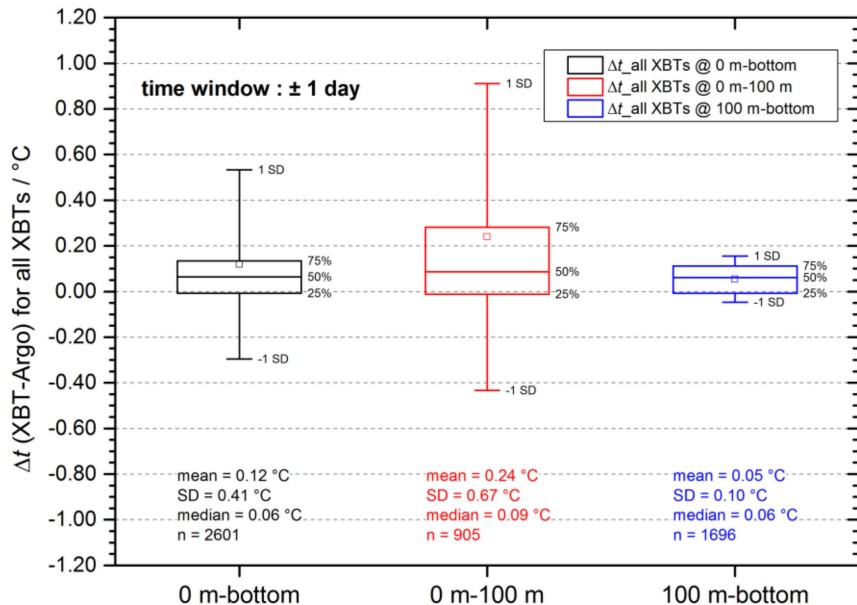

**Figure 9.** Box plot of temperature differences (XBT-Argo). Mean values are indicated by empty squares.

**Table 5.** Δ*t* (XBT-Argo) for each XBT type involved. Depth: 0 m-bottom (±1 day).

|  | T-4 & T-6 | T5 | Deep Blue |
|---|---|---|---|
| # matched points | 322 | 475 | 1804 |
| Max Δ*t* (°C) | 4.29 | 4.68 | 6.33 |
| Min Δ*t* (°C) | −0.43 | −0.84 | −1.55 |
| Mean Δ*t* (°C) | 0.12 | 0.09 | 0.13 |
| SD (°C) | 0.44 | 0.40 | 0.41 |
| Median Δ*t* (°C) | 0.07 | 0.01 | 0.07 |

**Table 6.** Δ*t* (XBT-Argo) for each XBT type involved. Depth: 0–100 m (±1 day).

|  | T-4 & T-6 | T5 | Deep Blue |
|---|---|---|---|
| # matched points | 128 | 143 | 635 |
| Max Δ*t* (°C) | 4.29 | 4.68 | 6.33 |
| Min Δ*t* (°C) | −0.37 | −0.84 | −1.55 |
| Mean Δ*t* (°C) | 0.18 | 0.27 | 0.24 |
| SD (°C) | 0.68 | 0.69 | 0.67 |
| Median Δ*t* (°C) | 0.05 | 0.06 | 0.10 |

**Table 7.** $\Delta t$ (XBT-Argo) for each XBT type involved. Depth: 100 m-bottom (±1 day).

| Quantity | T-4 & T-6 | T5 | Deep Blue |
|---|---|---|---|
| # matched points | 194 | 332 | 1169 |
| Max $\Delta t$ (°C) | 0.48 | 0.37 | 0.32 |
| Min $\Delta t$ (°C) | −0.43 | −0.16 | −0.49 |
| Mean $\Delta t$ (°C) | 0.07 | 0.01 | 0.06 |
| SD (°C) | 0.11 | 0.10 | 0.10 |
| Median $\Delta t$ (°C) | 0.08 | −0.01 | 0.07 |

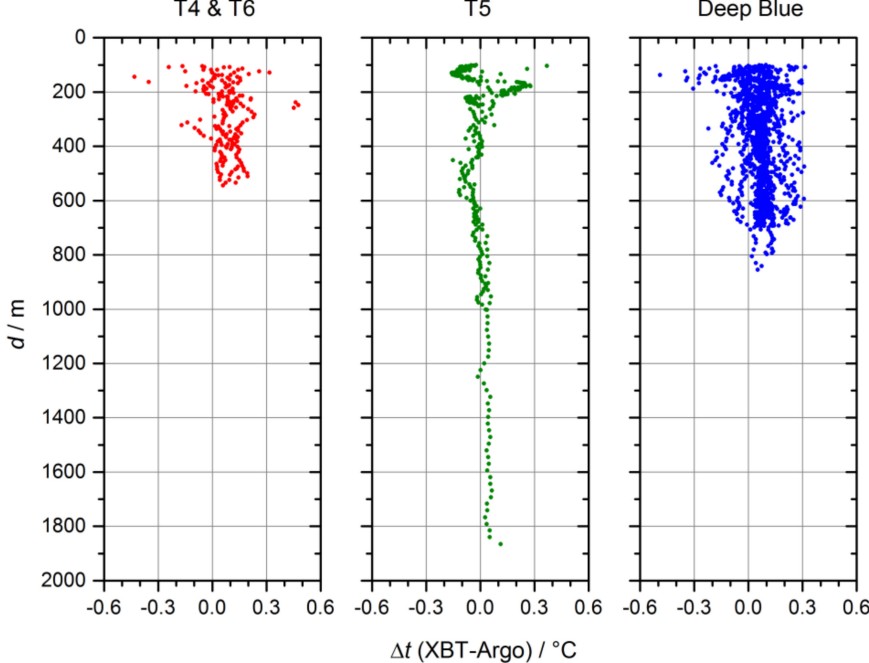

**Figure 10.** Temperature differences vs. depth ($d > 100$ m) for all the (XBT-Argo) pairs separated less than 10 km in space and within a 1-day time window.

Scatter diagram and linear regressions (1:1 line) were then applied on XBT vs. Argo values ($d > 100$ m): slope $a$ and the coefficient of determination $r^2$ showed no significant departure from the linearity (Figure 11).

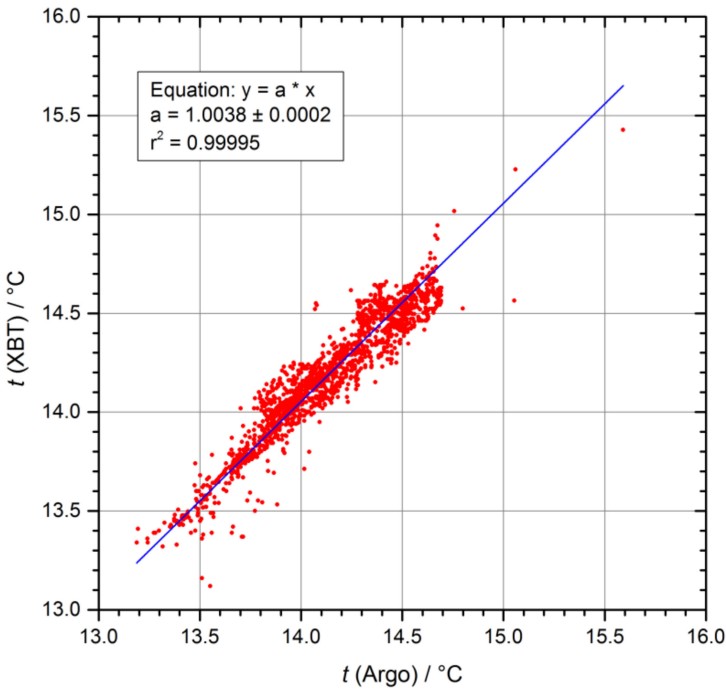

**Figure 11.** XBT vs. Argo values: linear fit (matched values for *d* > 100 m).

### 3.1.3. XBT vs. Argo Comparison: A Further Statistical Analysis

The population of (XBT-Argo) differences, exclusively associated with the smaller time window, was analyzed also by means of a paired sample *t*-test [39], used to determine whether the mean difference between two sets of observations can be taken as zero (i.e., the null hypothesis, meaning that the measures of the two instruments can be reasonably considered as equal to each other). The corresponding *t*-statistic is the ratio between the average of the differences between all pairs and the standard deviation of this average. Hence, the smaller this ratio, the more probable the null hypothesis is, which is instead rejected when the associated *p*-value is less than the significance level (0.05); *t*-tests were performed by means of R software [40]. Some obtained results, and their interpretation, can be summarized as follows.

By comparing T4 and T6 XBT measurements with Argo ones, in a strict statistical sense (i.e., neglecting the associated measurement uncertainties and applying the *t*-test of the pure data), the agreement is good, but just in the range 100–200 m (55 pairs, *p*-value equal to 0.24). This is due to a sufficiently large standard deviation of the differences with respect to the mean difference within that range, hence making the positive bias of XBT with respect to Argo measurements not statistically significant. Nonetheless, this result should be assessed also from a practical point of view: this agreement is satisfactory to the extent to which the corresponding amount of dispersion in the difference values is acceptable for measurement applications. For the T5 type, again without considering the instrumental uncertainties, the agreement is very good in the following two ranges: 200–400 m (83 pairs, *p*-value equal to 0.44) and 700–900 m (28 pairs, *p*-value equal to 0.33). The agreement between the two kinds of instruments in those ranges is clearly visible from Figure 10, where the differences are well centered on zero. In general, for T5, the statistical agreement with Argo is good all along the water column below 200 m (222 pairs, *p*-value equal to 0.64). Hence, it cannot be excluded that, in this case, the two instruments give the "same measure": this fact can be considered as a good indicator of the interchangeability of these two instruments, also indicating that, under these space-time conditions, the sea behaves reasonably like a thermostatic bath. This is enhanced by the seawater characteristics in the Mediterranean Sea (with a temperature range of about 1.0 °C even on 2–3 thousand meters of water) so that the temperature gradient is very small (frequently, some $10^{-3}$ °C·m$^{-1}$) making reasonable such

an expression. In addition, when the dense water formation occurs in winter months (e.g., in Gulf of Lyon or South Adriatic Sea), from surface down to about 500 m depth (or more), XBTs, Argo floats and CTDs are all able to measure a variation in temperature values not greater than 0.02–0.03 °C and this makes that example self-explanatory.

It has to be stressed, however, that the above-mentioned *t*-test was applied on pure data, considering them as perfectly known. Therefore, it resulted in being a very demanding comparison tool: when it is satisfied (and the variability in the differences is not too much large), it does indicate an actual strong agreement between the two kinds of instruments. However, when the test is not satisfied, it does not necessarily indicate an unsatisfactory agreement; rather, a proper metrological comparison should take into account also the measurement uncertainty involved in the process. The idea, in the present work, was to consider at least one of the two involved instruments with its associated standard uncertainty (if both the uncertainties were taken into consideration, an even better agreement would be certainly obtained).

Therefore, the normalized differences were calculated between the two instruments, i.e., |*t_XBT-t_Argo*|/*U*(*t_XBT*), where *U* is the expanded uncertainty associated with the measurement of the whole XBT system, and checked how many were found to be lower than one, hence indicating a satisfactory metrological agreement. For this purpose, neglecting the instrumental Argo uncertainty on temperature measurements (XBT temperature readings are intrinsically less accurate than Argo ones by a factor up to about 10 and similar conclusions for the depth sensors), a standard uncertainty of 0.1 °C was assigned to XBT measurements, obtained as a half of the overall XBT accuracy of 0.2 °C stated by the manufacturer [6,41]. This standard uncertainty can be considered as obtained "in field" (i.e., during working conditions in the sea, for a typical XBT launch from a traveling ship). Well, in this condition, XBT and Argo measurements were consistent also at depths in which, when neglecting uncertainty, the statistical test did not show a sufficient agreement. XBT and Argo could indeed be judged as metrologically consistent already from 100 m down. As a matter of fact, in this part of the water column, despite a "warm bias" in the XBT measurements of about +0.05 °C with respect to Argo values, there was at least a 90% proportion of normalized differences lower than 1. This result was valid as for all the XBTs as for each specific model (i.e., T4 and T6, T5 and Deep Blue). This means that the mean bias observed between XBT and Argo measurements is not significant for metrology applications and is in good agreement also with specific XBT vs. CTD in-field comparisons [42–44].

### 3.2. Argo vs. Ship-Based CTD Comparison

The matched Argo vs. CTD profiles were subdivided as follows:

- no. of matches with not-adjusted Argo profiles: 199 (with a total number *n* of matched points equal to 40,571 for both *t* and $S_P$ values);
- no. of matches with adjusted Argo profiles: 161 (with a total number *n* of matched points equal to 10,455 for both *t* and $S_P$ values).

It should be specified here that, as is reported in [33], the adjustment is a real value (i.e., decimal number) corresponding to the mean difference between original (real-time) and adjusted (delayed-mode) profiles of pressure, temperature or salinity for all values below 500 m depth. If a profile has an adjustment value, even if this value is 0.0, it indicates that the profile has gone through additional quality control by the Argo project and is considered either adjusted real-time or delayed-mode data. It has to be underlined that only in 75 pairs of matched profiles (of 161 with adjusted Argo values) an adjustment different from zero was reported; adjustments were applied only to salinity, with negative values ranging from −0.0018 to −0.0310 PSU.

In the following diagrams, an overview of Argo (adjusted and not-adjusted) and matched CTD profiles, together with main results obtained from the comparison, is reported.

Profiles of both *t* and $S_P$ vs. *d* were first plotted for all the Argo-CTD matched profiles, in order to have an overall indication of the space-time variability of the thermohaline properties in the area

(all Mediterranean Sea) and period of interest (about 12 years): diagrams are shown in Figures 12 and 13, respectively.

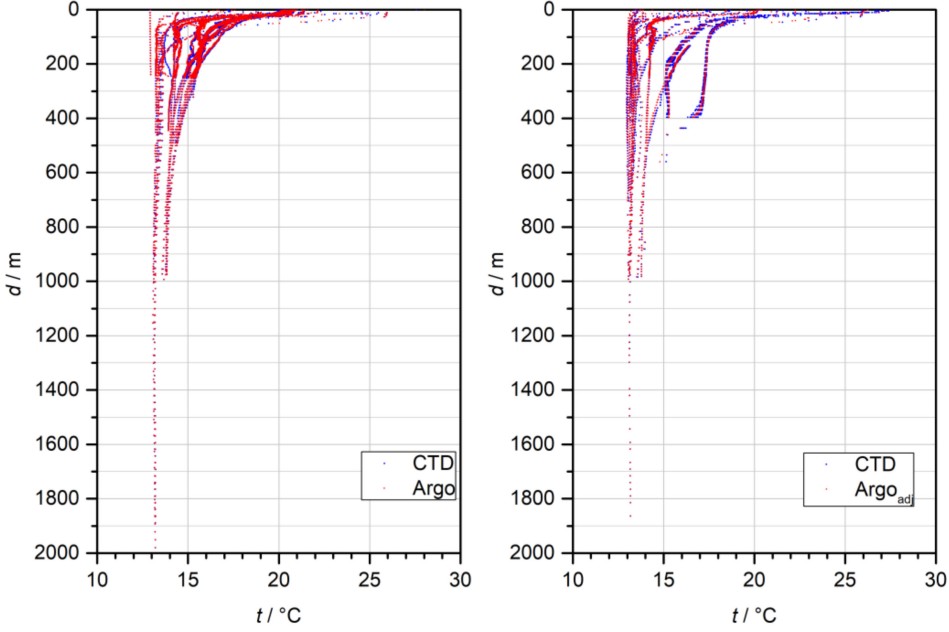

**Figure 12.** Overall Argo (red dots: not-adjusted, **left**, and adjusted, **right**) and CTD (blue dots) matched temperature profiles.

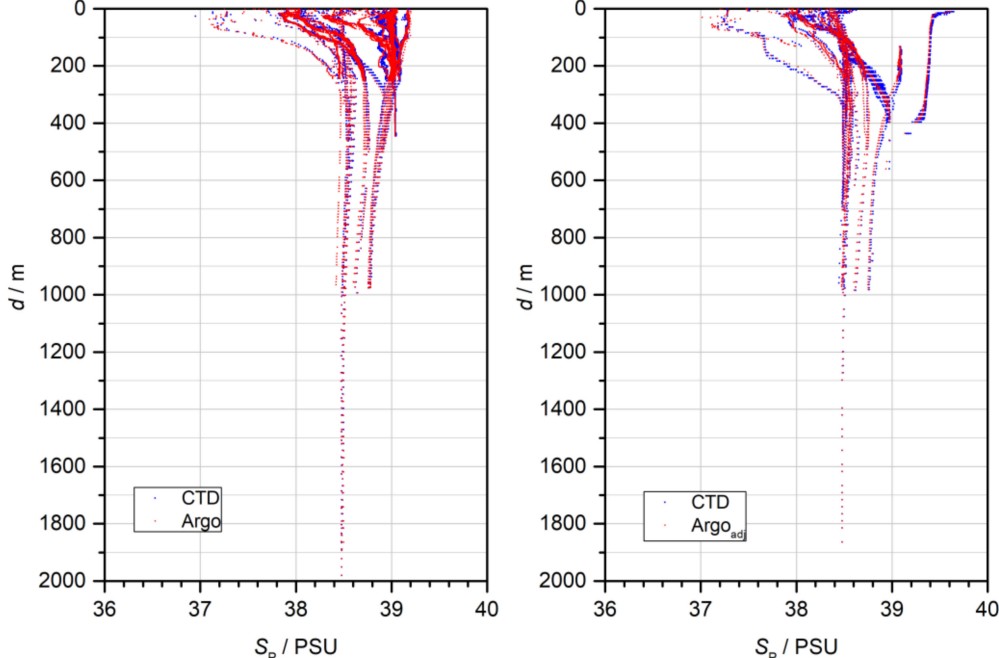

**Figure 13.** Overall Argo (red dots: not-adjusted, **left**, and adjusted, **right**) and CTD (blue dots) matched salinity profiles.

Values of $t$ and $S_P$ from the surface to 100 m depth were comprised substantially in the range of 13–28 °C and 37–39 PSU, respectively. At greater depths (i.e., $d > 500$ m), values converged in the interval 13.00–14.50 °C and 38.43–38.98 PSU, respectively.

All the paired values Argo vs. CTD in the matched profiles were then analyzed through box plots of *t* and $S_P$ differences (in analogy with Section 3.1.2): results are reported in Figures 14 and 15, respectively.

Considering the whole water column, temperature differences are negative on the average (mean equal to −0.02 °C) with SD equal to 0.20 °C in the case of not-adjusted Argo profiles. If adjusted Argo profiles are considered, even if on temperature a null adjustment was declared in the considered casts, a slight improvement in the mean difference can be noted, with a substantially identical dispersion: mean and SD values are in fact equal to 0.00 °C and 0.23 °C, respectively. For *d* > 100 m, mean differences are very close to zero, with SD values quite small (especially again in the case of adjusted values). This fact can reasonably imply that data on which an additional quality control is not applied may be subjected to a potential systematic error.

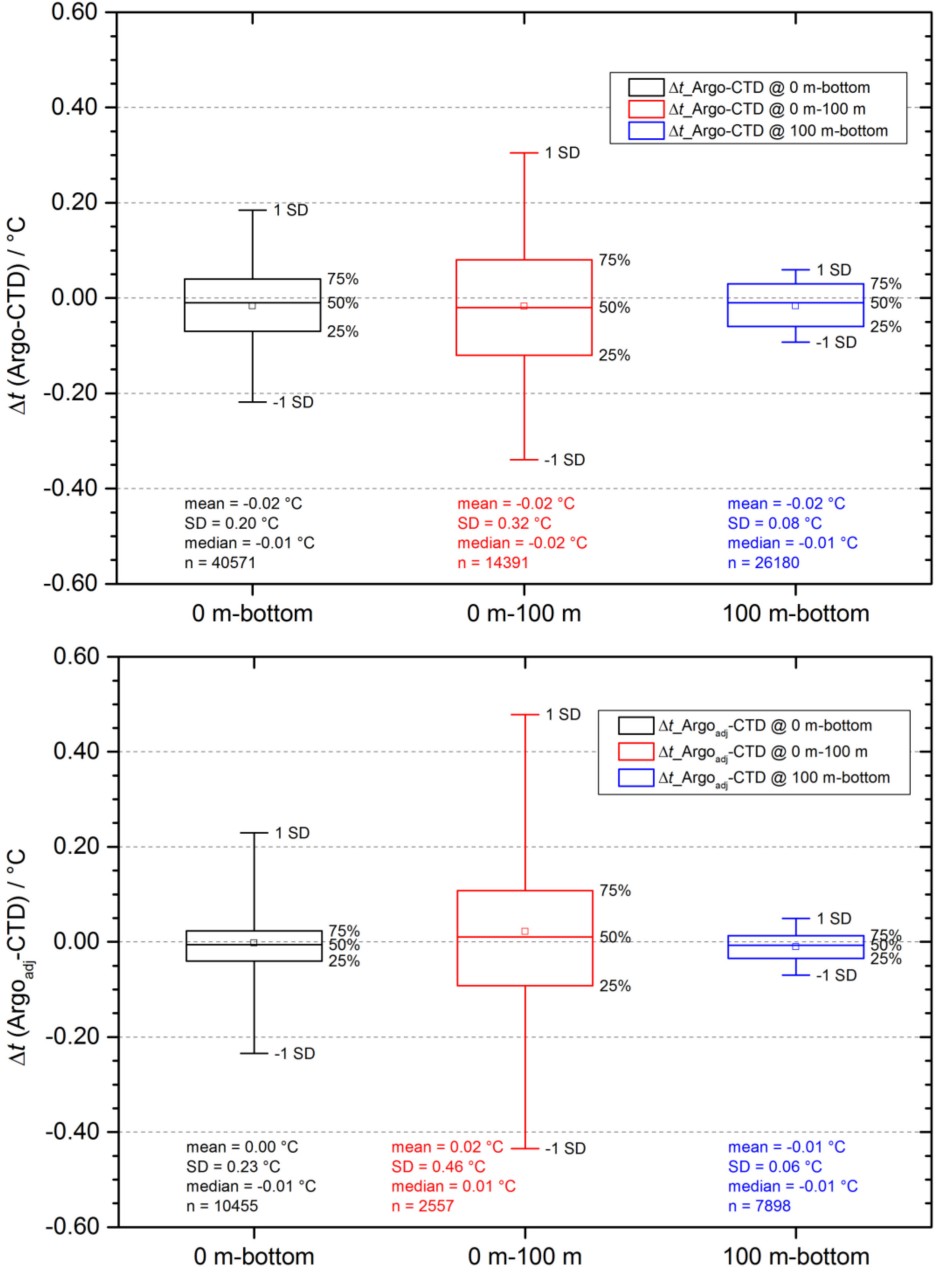

**Figure 14.** Box plot of temperature differences: Argo (not-adjusted, **top**, and adjusted, **bottom**) vs. CTD. Mean values are indicated by empty squares.

For the salinity data, again, a slight improvement in the mean difference between Argo vs. CTD can be noted when adjusted Argo data are considered; if all the water column is considered, mean values change in fact from −0.013 PSU to −0.009 PSU (with SD equal to 0.041 PSU and 0.038 PSU, respectively). For $d > 100$ m, mean differences are again reduced (−0.009 PSU vs. −0.011 PSU, with a smaller SD).

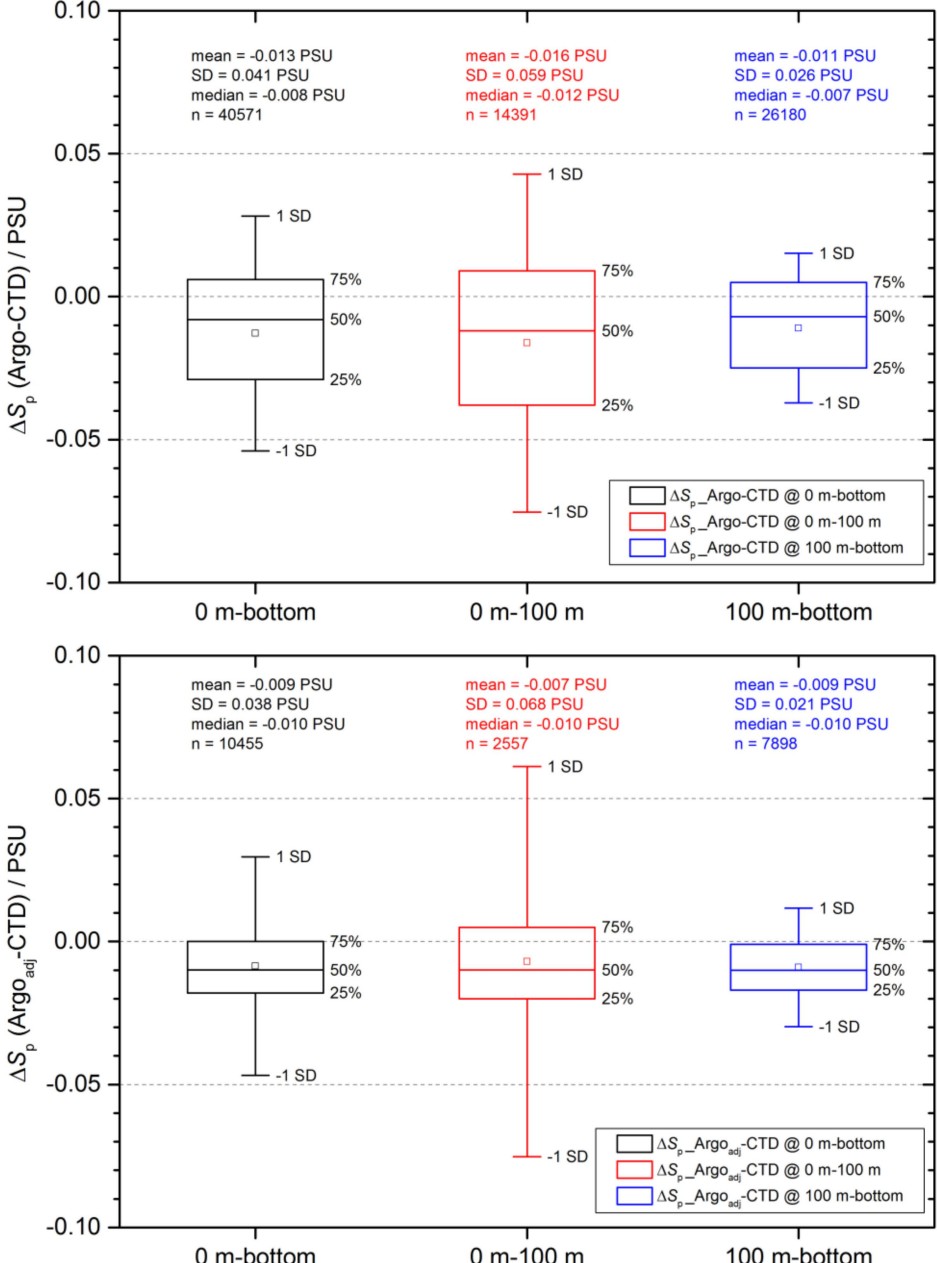

**Figure 15.** Box plot of salinity differences: Argo (not-adjusted, **top**, and adjusted, **bottom**) vs. CTD. Mean values are indicated by empty squares.

Matched profiles were then sorted in depth intervals distributed along the water column (at 100 m step down to 1000 m, then a single step at deeper depths down to 2000 m); for each depth interval, mean and SD of both $t$ and $S_P$ differences were calculated. Results are plotted in Figures 16 and 17, where $n$ in this case indicates the number of considered pairs in each interval.

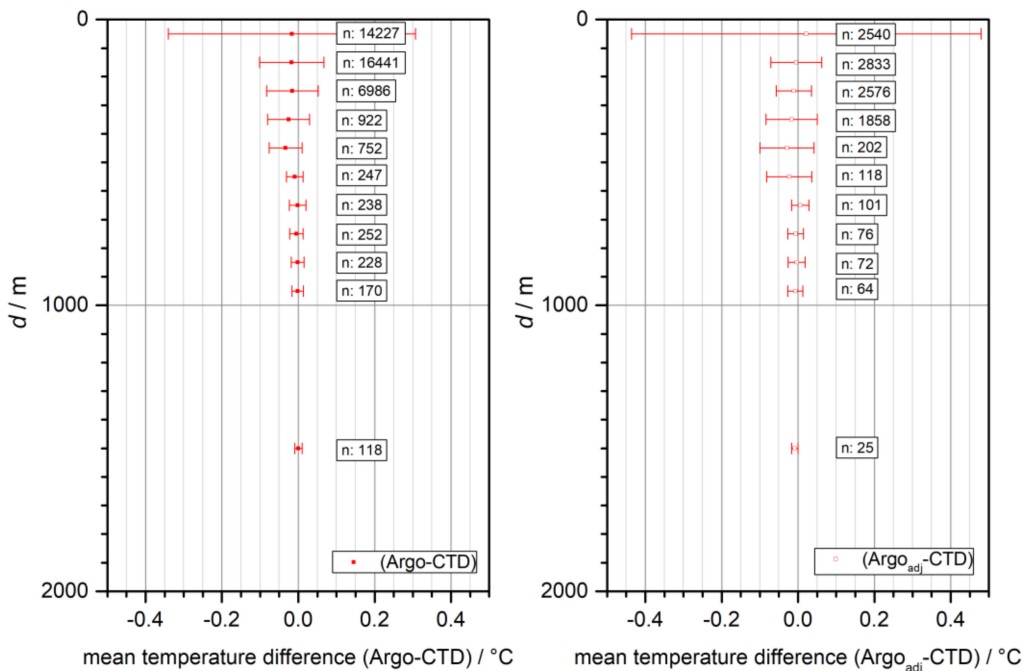

**Figure 16.** Mean temperature differences and SD at depth intervals: Argo (not-adjusted, **left**, and adjusted, **right**) vs. CTD.

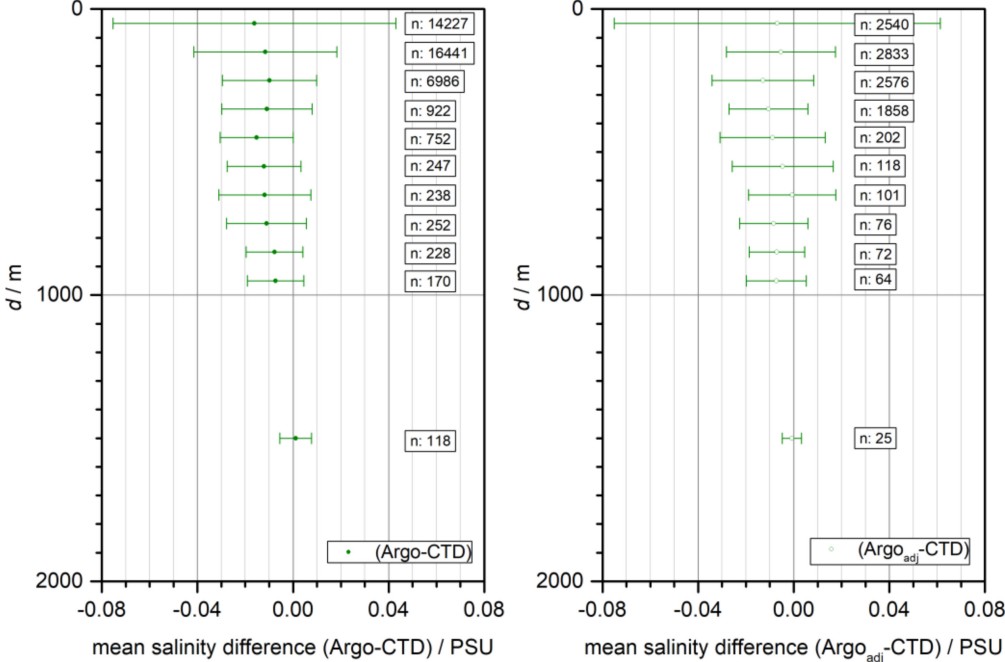

**Figure 17.** Mean salinity differences and SD at depth intervals: Argo (not-adjusted, **left**, and adjusted, **right**) vs. CTD.

The results show that the temperature difference (Argo vs. CTD, in both cases), averaged in the intervals, spans in the range −0.03 to +0.02 °C in the layers down to 500 m. From 600 m down, this difference becomes less equal to −0.01 °C. The SD values substantially decrease toward deeper depths, due to the reduction of sea water temperature variability with depth itself; values span from about 0.4 °C near the surface to about 0.01 °C at the bottom of the profiles. In terms of salinity, the mean differences from the surface down to 500 m have values ranging from −0.016 to −0.005 PSU. Deeper in

the water column, the differences typically converge to −0.005 PSU. Values of SD, again, are greater near the surface (up to about 0.070 PSU), reaching a minimum at bottom depth (about 0.010 PSU).

Scatter diagrams and linear regressions (1:1 line) were then applied on both Argo $t$ and $S_P$ values vs. the matched values obtained with ship-based CTDs: results are shown in Figures 18 and 19, respectively. The slope $a$ of the regression model shows again a slight improvement in the one-to-one relationship for adjusted Argo values (for both $t$ and $S_P$); in general, no significant departure from the linearity was observed for any of the two quantities under study. Due to the fact that both Argo and ship-based CTDs host similar sensors, a very strong linearity and high linear correlation should be reasonably expected, even in cases when there is an actual bias between the profiles.

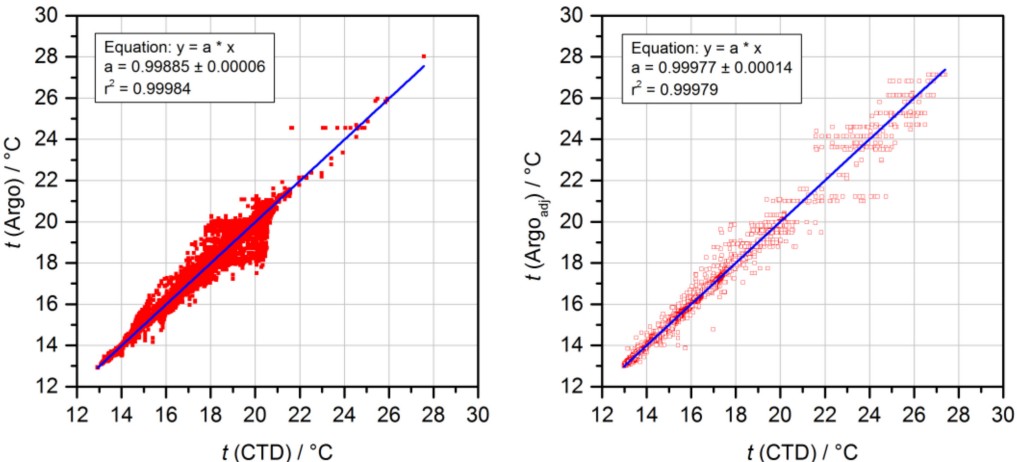

**Figure 18.** Scatter diagram and linear regression (slope $a$, coefficient of determination $r^2$) of Argo (not-adjusted, **left**, and adjusted, **right**) vs. CTD values: temperature (all data in the water column).

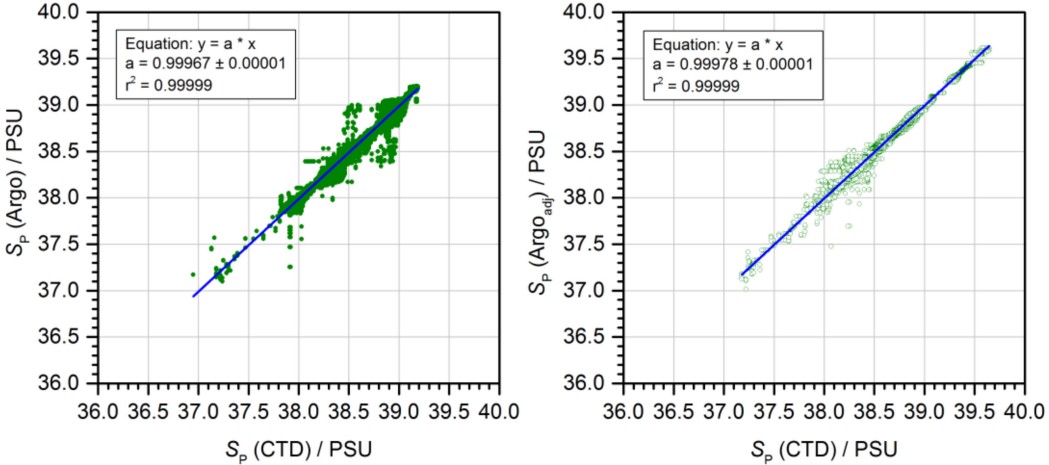

**Figure 19.** Scatter diagram and linear regression (slope $a$, coefficient of determination $r^2$) of Argo (not-adjusted, **left**, and adjusted, **right**) vs. CTD values: salinity (all data in the water column).

Finally, for each involved Argo float, mean differences vs. CTD values, for $d > 100$ m, were plotted as a function of the time elapsed since each float deployment, in order to give evidence of possible drifts of overall Argo population. Diagrams are reported in Figure 20.

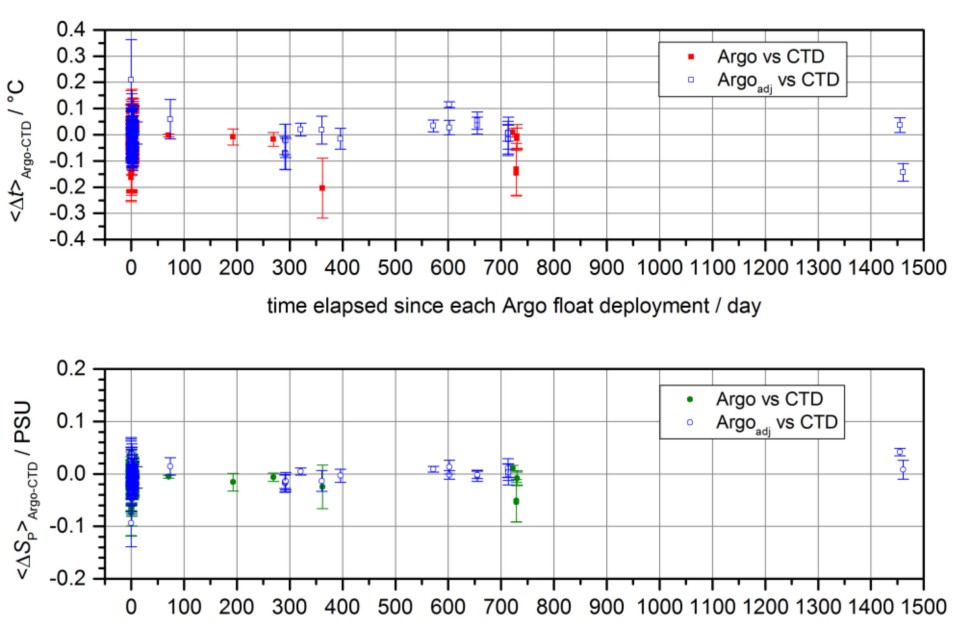

**Figure 20.** Argo (not-adjusted and adjusted) vs. CTD: temperature (**top**) and practical salinity (**bottom**) mean differences (indicated by symbols < >) under 100 m plotted vs. time elapsed since Argo deployment.

### 3.2.1. Argo vs. Ship-Based CTD Comparison: A Further Statistical Analysis

Temperature and salinity differences (Argo-CTD) were analyzed in terms of *t*-test as in Section 3.1.3. For temperature values, by comparing not-adjusted Argo and CTD measures, *t*-test results showed that at $d > 800$ m the two instruments gave the "same measure", in a strict statistical sense, that is when neglecting the associated measurement uncertainties (with a slight tendency of the CTD measurements to be greater than the Argo ones, being the mean difference at these depths equal to about −0.001 °C). On the contrary, by comparing adjusted Argo and CTD measures, again without considering their instrumental uncertainties, the null hypothesis had to be rejected along the whole water column (i.e., there was no evidence of equal behavior, in a strictly statistical sense, between the two instruments). Anyway, for $d > 800$ m, the mean temperature difference (Argo$_{adj}$ -CTD) was about equal to −0.006 °C, that can be considered as a good indicator of the interchangeability of these two instruments. The very stringent results of the *t*-test, applied to Argo and CTD measurements, were due to the observed low dispersion in the differences (about half of the SD of the differences between XBT and Argo measurements on the whole water column) which made statistically significant even a small mean difference, that is "enough different" from zero. As an example, the mean of (not-adjusted) temperature differences in the region 500–600 m was −0.007 °C (Figure 16, left panel), whereas the corresponding standard deviation (equal to the population SD divided by the square root of 247, the sample size) was equal to 0.001 °C, much smaller than the absolute mean bias (0.007), hence leading to a significant *p*-value (7E-08). Therefore, although the (absolute) mean difference is quite small, it resulted in being statistically different from zero, from the point of view of the *t*-test. Considering such a situation, the fact that under 800 m the not-adjusted Argo and CTD measurements passed the test indicates that, in this zone of the water column, the two instruments were in very good agreement and the sea behaved reasonably like a thermostatic bath (the natural variability of the thermohaline properties is reduced, [35]). Moreover, as performed for the differences between XBT and Argo, a full metrological comparison was needed, taking into account the standard uncertainty associated with at least one of the two instruments, as obtained "in field" (i.e., during working conditions in the sea,

in a typical ship-performed cast). For this purpose, neglecting the instrumental Argo uncertainty on temperature, the standard uncertainty of 0.023 °C was assigned to CTD measures, as reported in [45]. In this condition, Argo (both not-adjusted and adjusted) and CTD measures were metrologically consistent also at depth intervals in which, when neglecting uncertainty, the statistical test did not show a sufficient agreement. Argo and CTD, within the considered uncertainty, could be considered as measuring the same quantity already from 500 m down, in the sense that, in this water column, there is at least a 90% percentage of normalized differences (i.e., $|t\_Argo-t\_CTD|/U(t\_CTD)$) lower than one. For $d < 500$ m, mean differences are as large as −0.02 °C, but with greater SD due to the natural variability of sea temperature towards surface layers. Therefore, any significant offset between Argo and ship-based CTD can hardly be identified, due to the fact that mean differences are of the same magnitude order as the standard uncertainties of instruments in field.

Considering salinity measurements, following the same criterion (i.e., neglecting instrumental uncertainties), not-adjusted Argo salinity measurements were compared with CTD ones. The null hypothesis was accepted only for $d > 1000$ m; from surface to 1000 m the mean difference was about −0.01 PSU. The Delayed Mode Quality Control caused a slight improvement: for adjusted Argo values, in fact, the statistical agreement under 1000 m was stronger (i.e., showing higher *p*-values) and a good agreement was reached also in the range 600–700 m. From surface to about 1000 m depth, the mean difference was lowered to about −0.007 PSU. By considering again the CTD uncertainty in field for salinity, equal to 0.01 PSU as reported in [45], not-adjusted Argo and CTD can be considered as reasonably measuring the same quantity already from about 500 m depth. For adjusted Argo data, a good agreement (in the sense of metrological data consistency) was reached only at $d > 800$ m. It can in any case be concluded that also for salinity any significant offset between Argo and ship-based CTD would be masked by the transducer standard uncertainty.

Finally, for what concerns the Argo float stability, a Weighted Least Squares linear fit was applied to data reported in Figure 20 [46]: the purpose was to assess if the parameter *b* (slope) of the model $y = a + b·x$ (where *x* indicates the time, expressed in days), is significantly different from zero (i.e., indicating the presence of drift) or not (i.e., absence of drift). The slope is different from zero when its standard error is small compared to the numeric value $|b|$, and consequently the associated *t*-test gives a *p*-value smaller than 0.05 (i.e., the null hypothesis of a slope equal to zero cannot be accepted). Results are shown in Table 8.

**Table 8.** Statistical estimates of Argo drifts on temperature and salinity measures.

| Argo Data | Slope *b* | *t*-Test |
|---|---|---|
| not-adj_temp | (−0.6 ± 1.4)E−05 °C/day | NO drift |
| yes-adj_temp | (6.2 ± 1.0)E−05 °C/day | YES drift |
| not-adj_sal | (21 ± 9)E−06 PSU/day | YES drift |
| yes-adj_sal | (29 ± 2)E−06 PSU/day | YES drift |

## 4. Discussion

The first aim of the present work was to assess the temperature difference between XBT probes and Argo profiling floats, quasi-collocated and quasi-simultaneous along the SOOP Genoa-to-Palermo transect, in a period of about 15 years (up to March 2019) [47,48]. Some considerations should be set out here. First of all, the depth of an XBT probe is not measured directly but is estimated through a fall rate equation with empirical coefficients (based on tests carried out by the manufacturer [32]) which change with the XBT type but which are independent of any other factor such as water temperature, launching height and so on. On the other hand, a value as large as 0.2 °C is proposed by manufacturers as the overall accuracy on temperature reading of an XBT system, which consists of the XBT probe itself and the recording system. Without additional information, it is impossible to separate the specific contribution due to the specific probe and the used recording system. Many phenomena have been

identified that can contribute to the uncertainties in the measurements performed by XBTs, and they can act either on the depth or on the temperature or on both [14]. The solution to the problem of how to correct the values measured by XBT probes has not yet been found [43,44]. For this reason, no corrections have been applied to the depth values of the XBTs in this work. Therefore, the results obtained herein (which are still affected by an error on the read $t$ value, attributable to an incorrect $d$ value calculated by the fall equation) could certainly be improved, in terms of temperature mean difference and standard deviation. Furthermore, it has been verified that, in a significant part of available comparisons, the calculated depth for the XBT overestimates the actual depth in the first tens of meters of the fall. This is well evident when significant thermal structures occur (i.e., the start of the summer thermocline): the temperature actually measured is combined with a calculated depth deeper than the real value. For any depth value up to 250 m, the associated uncertainty is 5 m. This value does not significantly affect the temperature variation measured at greater depths. On the other hand, in the 0–100 m depth region it can have even heavy consequences where the structures of the upper thermocline start, and gradients greater than 2 °C·m$^{-1}$ can be measured. In the winter period the gradient is much smaller and XBTs are able to better describe the local water temperature, so that their bias and SD are homogeneous along the whole water column. An example of this can be verified in Figure 7, with temperature differences up to about +8 °C. Therefore, in the surface and sub-surface layers a small variation in depth could in fact correspond to a very important variation in temperature, so that the comparison with Argo values would not be significant. We also note that daily variability due to solar radiations and the occurrence of strong winds could reduce the significance of measurements in near-the-surface region even within a 1-day constraint, also because of a possible contribution of internal waves. That said, in this work, a first-time window of ± 7 days was initially chosen to build a first, conspicuous database of (XBT-Argo) pairs. Then, a more strict matching condition in terms of time was applied, by choosing a window of ± 1 day (with the same space matching conditions, equal to about 10 km). Results in terms of $t$ mean difference, calculated on all XBTs, are not significantly dissimilar in the two considered time windows: for $d > 100$ m, in fact, values of 0.04 °C and 0.05 °C were found, respectively (with SD values equal to 0.12 °C and 0.10 °C). Further statistical analysis was then applied to temperature differences related to different XBT type involved in this comparison. Application of the paired sample $t$-test showed a general superiority of the T5 model with respect to the other XBT types. However, a full metrological comparison between XBT and Argo measurements, taking into account the XBT measurement uncertainty, proved a good behavior of all kinds of XBT at all water depths, from 100 m down. In summary, XBT temperature profiles, collected in the Western Mediterranean Sea by commercial vessels, have proven not to differ too much, from a metrological point of view, from the values recorded by Argo profilers, considering position differences smaller or similar to the local Rossby radius. By varying the time windows of the comparison from daily to weekly scales, differences do not change significantly. If the complete profile of the different types of XBTs usually launched is considered, the agreement between the values recorded by Argo and XBTs is poor. However, if the near-surface region is eliminated (usually identified in the 0–100 m layer, which is critical for the XBTs and where generally significant thermal structures are present), the agreement in the lower layers becomes much more consistent. In summary (Table 9), temperature values provided by XBTs in the deeper region show a slight excess (+0.05 °C) that, combined with its SD (0.10 °C), is fully compatible with the accuracy declared by the manufacturers. The application of a more accurate description of the falling motion of the XBT probes and a more accurate evaluation of factors acting on the thermal component of XBT measurements should consequently allow a reduction (but small) in the differences between the reading by XBT and Argo profilers.

The second aim of the present work was to assess the differences in both temperature and salinity between Argo profiling floats and ship-based CTD casts quasi-collocated and quasi-simultaneous, in all the Mediterranean Sea and in a period of about 12 years (up to June 2018). Strict space-time matching conditions allowed us to build a conspicuous database of Argo-CTD pairs. Obtained conclusions are summarized in Table 9. It should be underlined that for both $t$ and $S_P$ values, mean differences

(Argo-CTD) are overall negative, in agreement with [35]; furthermore, for both $t$ and $S_P$ values, an accuracy improvement can be noted (in terms of mean values and dispersion reduction) due to the adjustment method. For $d > 100$ m the biases of Argo vs. CTD are of the order of about 0.01 °C and 0.01 PSU, respectively. Concerning the statistical analysis on the temperature differences, application of the (very strict) paired sample $t$-test was satisfactory only for $d > 800$ m, but the metrological comparison showed a good agreement between the two instruments for $d > 500$ m. In the upper layers, any significant offset between Argo and ship-based CTD can hardly be identified, because their mean differences are of the same magnitude order of the standard uncertainties of instruments in field. Furthermore, concerning salinity differences, the statistical agreement on pure data was reached at high depths (generally at $d > 1000$ m), but the metrological agreement was proved for $d > 500$ m and $d > 800$ m, for not-adjusted and adjusted Argo measurements, respectively. It was concluded that, also for salinity, any significant offset between Argo and ship-based CTD would be masked by the transducer standard uncertainty. Finally, Argo drift during its lifetime was analyzed by available experimental data; results showed that even adjusted data are affected by drift, but its effect is confirmed as negligible (i.e., reasonably comparable with standard uncertainty) when considered within the mean lifetime of an Argo float (about 4 years).

**Table 9.** Final assessment of biases (with associated SD).

| Inter Comparison | Bias on Water Column | | Bias for $d > 100$ m | |
|---|---|---|---|---|
| | $t$ / °C | $S_p$ / PSU | $t$ / °C | $S_p$ / PSU |
| **XBT vs. Argo** | 0.12 ± 0.41 | | 0.05 ± 0.10 | |
| **Argo vs. CTD** | −0.02 ± 0.20 | −0.013 ± 0.041 | −0.02 ± 0.08 | −0.011 ± 0.026 |
| **Argo_adj vs. CTD** | 0.00 ± 0.23 | −0.009 ± 0.038 | −0.01 ± 0.06 | −0.009 ± 0.021 |

In conclusion, in order to homogenize data processing, both transducer users and scientists involved in ocean modeling are encouraged to take into account bias results, reported in Table 9, when managing temperature and/or salinity data profiles acquired by XBT, Argo floats and ship-based CTDs.

**Author Contributions:** Conceptualization, F.R., G.R. and F.P.; methodology and formal Analysis, G.R., F.P. and F.R.; XBT data acquisition, F.R.; contribution to dataset managing, A.B. and L.R.; funding acquisition, F.P. All the authors discussed the results and contributed to the manuscript writing. All authors have read and agreed to the published version of the manuscript.

**Funding:** The APC was funded by Istituto Nazionale di Ricerca Metrologica (INRIM), www.inrim.it.

**Acknowledgments:** (a) Part of the XBT probes deployed after 2009 were kindly provided by AOML/NOAA in the framework of JCOMM/SOOP activities. (b) Many thanks to the following shipping companies: Grandi Navi Veloci (GNV), for the period 1999–2019; Hapag Lloyd, for the period 2007–2010; CMA CGM, for the period 2009–2010; Arkas Lines, for the period 2013–2014. (c) The Argo profiles in the Mediterranean were collected and made freely available (http://doi.org/10.17882/42182) by the International Argo Program (http://www.argo.ucsd.edu, http://argo.jcommops.org), in particular the Coriolis project and programs that contribute to it (http://www.coriolis.eu.org). The Argo Program is part of the Global Ocean Observing System. (d) The authors wish to thank the SeaDataCloud project (European Union's Horizon 2020 Research Infrastructures Programme, grant agreement No. 730960) that allowed participation to 2019 IMEKO TC-19 International Workshop on METROLOGY FOR THE SEA (October 3–5, 2019, Genova, Italy). (e) The authors wish also to thank the reviewers for their help in the improvement of the quality of the paper.

**Conflicts of Interest:** The authors declare no conflict of interest.

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
