# Peer review of "XBT, ARGO Float and Ship-Based CTD Profiles Intercompared under Strict Space-Time Conditions in the Mediterranean Sea: Assessment of Metrological Comparability"

_jmse, doi:10.3390/jmse8050313_

Round 1

Reviewer 1 Report

< Specific comments>

The manuscript "XBT, ARGO float and ship-based CTD profiles intercompared under strict space-time conditions in the Mediterranean Sea: assessment of metrological comparability" is within the scope of Journal of Marine Science and Engineering.

This research aim of the present work was also to show the main results obtained from comparing (adjusted and not-adjusted) Argo profiles vs. ship-based CTD profiles, mated under strict space and time matching conditions. Simultaneously, the author attempts to fill the knowledge gap of past literature with the results of this research.

This issue is a very increasingly important issue in oceanography or changes in the marine environment (such as water temperature or salinity). Especially, the seasonal change of ocean currents and Thermocline. In the manuscript, the data and methods are thoughtful and relatively well presented. But, some content needed a minor adjustment. And, there are several key problems the author still needs to address. Especially, the author can provide what specific recommendations for future CTD, ARGO, XBT users or research through the results of this study.

  1. Abstract: The author did not provide enough information in the abstract, for example, research purposes are insufficient. And, the specific research discussions and recommendations on future CTD, ARGO, XBT users or research.
  2. Introduction: Include in the introduction the current/existing debate in the field/topic, hence, why the paper is warranted; and the knowledge gap that the paper is trying to fill. Such statements will establish the reason behind the research and its significance.
  3. Discussion and future research recommendations:(1) How would the evidence presented in the manuscript be an important contribution to literature and would how can they have potential use for other similar study types or study areas?(2) What specific suggestions can this manuscript content provide according to the research result of the Mediterranean Sea for international audiences?  Especially, the future users or research of CTD, ARGO, XBT.
  4. References: Are all the references and literature citations complete, accurate and consistent with journal style? Please check.

Author Response

All row references are referred to revised version of the manuscript .

Some minor changes (e.g. captions, axis labels) have been applied in the text (all changes are traceable by Track Change Tool in Word) 

*********************************************************

Request - Abstract: The author did not provide enough information in the abstract, for example, research purposes are insufficient. And, the specific research discussions and recommendations on future CTD, ARGO, XBT users or research.

Reply - Two sentences were added/modified  to the Abstract:

1. rows 16-20: Accurate measurement of temperature and salinity is a fundamental task with heavy implications in all the possible applications of the currently available datasets, for examples in the study of climate changes and modeling of Ocean dynamics. In this work, the reliability of measurements obtained by oceanographic devices (XBTs, Argo floats and CTDs) is analyzed by means of an intercomparison exercise.

2. rows 37-40: XBT, Argo and CTD users are therefore encouraged to take into account these results as a good indicator of the uncertainties associated to such devices in the Mediterranean Sea, for the analyzed period, in all the climatological applications. 

*********************************************************

Request - Introduction: Include in the introduction the current/existing debate in the field/topic, hence, why the paper is warranted; and the knowledge gap that the paper is trying to fill. Such statements will establish the reason behind the research and its significance.

Reply - One sentence was added to the Introduction:

rows 50-53: The main purpose was to assess the metrological comparability of such transducers (in about the last two decades) in the Mediterranean Sea, which is a marginal sea with both unusually high temperature and salinity values and peculiar shape of their profiles.

.

*********************************************************

Request - Discussion and future research recommendations:(1) How would the evidence presented in the manuscript be an important contribution to literature and would how can they have potential use for other similar study types or study areas? (2) What specific suggestions can this manuscript content provide according to the research result of the Mediterranean Sea for international audiences? Especially, the future users or research of CTD, ARGO, XBT.

Reply- One sentence was added to the Discussion and a final table (no.9) summarizes the results describing the sensor perfomances in the Mediterranean Sea.

  • row 616-619: In conclusion, in order to homogenize data processing, both transducer users and scientist involved in Ocean modeling are encouraged to take into account bias results, reported in Table 9, when managing temperature and/or salinity data profiles acquired by XBT, Argo floats and ship-based CTDs.     
  • row 620 (Table 9)

*********************************************************

Request-References: Are all the references and literature citations complete, accurate and consistent with journal style? Please check.

Reply-References checked, minor changes (punctuation, spaces) in References.

Now, we have 48 references (added no. 43 and 44)

*********************************************************

Reviewer 2 Report

Overall this is a strong paper presenting a useful analysis. Most of the text is concise and the results are generally well presented.

Problems arise, however, in the Discussion section. Additional and critical statistical analyses are presented in the “Discussion”, and these statistical tests and measures properly belong back in the “Results” section. These additional results need to be displayed more systematically with tables so that all of the statistical pair-wise t-test results may be seen between each instrument / depth-interval / variable/ or however the authors have chosen to parse these data.

Apparent disagreements between the earlier analysis and the “Discussion” statistical tests (t-tests) need to be much more thoroughly explained.

For example, Figure 10 clearly shows that T4, T6, and Deep Blue XBT instruments have a consistent positive bias over the Argo in the > 100m depth interval, i.e., these XBT instruments are consistently measuring higher temperatures than Argo. Yet, the t-tests indicate some agreement over selected depth intervals – it is difficult for the me to reconcile these analyses (bulk delta t versus t-tests) without a more systematic / thorough explanation and display of the complete paired t-test results. The results section clearly demonstrates there is an inherent problem (warm bias) with XBT temperatures compared to Argo.

Lines 447-459 – Again, perhaps authors should present this more methodically in the Results section and explain the choices more carefully. For example, why are the authors dividing measurement uncertainty by 2?

Lines 459-472 – I respectfully disagree with the conclusions presented herein: it looks to me that XBT (except T5) is consistently overestimating temperature, if we presume that Argo is closer to truth, ( see in Figure 10, Tables 5-7). What is causing this consistent positive bias for XBT? Is this a depth calculation problem for XBT? Can it be resolved/altered with a depth calculation correction?  

Lines 480-502: Here again, more explanation of the t-test results is needed. Let’s go back to Figure 14 – the histogram of pair-wise differences between Argo and CTD temperatures – it looks fairly “normal” or “gaussian” around a mean of differences very close to 0 (left panel) and exactly 0 (right panel). This leads me to believe that the Argo vs CTD temperature comparisons are very good - with much of the remaining dispersion simply due to unconstrained time/space mismatch – hence the histogram’s gaussian symmetry. Under the presumption of a normal probability density function, and the histogram looks normal, then in both cases (left and right) ~ 95% of the Argo v CTD temperature differences are within ~ +/- 0.4 degrees C. That strikes me as good agreement considering inherent imperfections in space/time match-ups within one Rossby radius of deformation.  

Yet, the t-test leads authors to reject the null hypothesis for nearly all of the Argo vs CTD comparisons (line 487). Are the authors concluding the Argo/CTD agreement is poor based strictly on pair-wise t-test p-values? Argo v CTD (t) appears much closer than XBT v Argo (t), and that appears at odds with the mixed t-test results as they are presented in the discussion.

Are the mean differences in Figure 16 for each depth interval larger or smaller than the delta (t) shown in the tables (5-7) for XBT v Argo ? And why not show histograms of XBT v Argo differences as well? Some uniformity/consistency of comparison graphs for the 2 sets of comparisons (XBT v Argo, and Argo v CTD) would help.

In summary – the t-tests and p-values stuffed into the Discussion section created confusion for me and appear at odds with more obvious conclusions that would follow from the earlier analysis presented in the results. If authors apply the same due diligence to the t-tests as they obviously have with the remaining bulk of the analysis, I’m sure this can be easily corrected. If t-tests reach different conclusions for XBT v Argo than Argo v CTD, these conclusions need to be reconciled with the preceding analysis.

The last section on Argo float stability should also be in the results section. Final paragraph/section of the paper should provide an overall summary of the important findings and their implications.

Additional Comments:

Strong linearity and very high linear correlation for measurement agreement (For example, Figures 18 and 19) are expected from ocean profile data (especially, temperature/density). For example, 2 temperature profiles could be apart by 10 degrees or more, and still have a nearly perfect linear correlation since the bulk profile variance is determined by depth. Authors might consider, in the future, a different metric of measurement agreement dispersion such as Root-Mean-Square-Difference.  

Authors repeatedly make references to the ocean as a “thermostatic bath.” I do not follow this comparison. If deep ocean temperature does not change, then why are we measuring it?

Additional suggestion: upper ocean variance in temperature may be due to diurnal surface warming during boreal spring/summer. Would a seasonal parsing of these data improve any < 100 m comparisons?

This is a well-written and organized paper overall. Some revision of the t-test results and their explanation in the revision should be sufficient for potential publication.

Minor detail: lats/lons on Figure 2 are in very small font – might want to reduce tick interval and enlarge font.

Author Response

All row references are referred to revised version of the manuscript .

Some minor changes (e.g. captions, axis labels) have been applied in the text (all changes are traceable by Track Change Tool in Word) 

*********************************************************Request - Additional and critical statistical analyses are presented in the “Discussion”, and these statistical tests and measures properly belong back in the “Results” section.

Reply - Statistical analysis have been moved back to Result Section, by adding two proper sub-sections (3.1.3. and 3.2.1.)

*********************************************************

Request - These additional results need to be displayed more systematically with tables so that all of the statistical pair-wise t-test results may be seen between each instrument / depth-interval / variable/ or however the authors have chosen to parse these data.

Reply - t-test results, in terms of p-values, need to be fully explained by means of some further considerations: so, we have decided to discuss them in proper sub-sections (3.1.3. and 3.2.1.) without any tables (results shown in tables, in fact, aren't  self explanatory). Furthermore, in our opinion, only t-test results related to the whole water column (or at least below the thermocline region, for depth greater than 100 m) provide good information on the performances of the analyzed instruments for metrological intercomparison.

*********************************************************

Request - Apparent disagreements between the earlier analysis and the “Discussion” statistical tests (t-tests) need to be much more thoroughly explained. For example, Figure 10 clearly shows that T4, T6, and Deep Blue XBT instruments have a consistent positive bias over the Argo in the > 100m depth interval, i.e., these XBT instruments are consistently measuring higher temperatures than Argo.

Reply - Rows 263/267 and Tables from 2 to 7: the mean XBT-Argo temperature difference  seems to be consistent, but actually the average positive bias of XBT measurements with respect to Argo is encompassed within the data standard deviation.

*********************************************************

Request - Yet the t-tests indicate some agreement over selected depth intervals – it is difficult for the me to reconcile these analyses (bulk delta t versus t-tests) without a more systematic / thorough explanation and display of the complete paired t-test results. The results section clearly demonstrates there is an
inherent problem (warm bias) with XBT temperatures compared to Argo.

Reply - t-tests results are now more clearly explained in proper sections  (3.1.3. and 3.2.1.). Moreover, t-tests have been applied to pure data, without considering the measurement uncertainty: a good t-test result does imply a very strong metrological comparability between the compared instruments. On the other hand, the very small dispersion of the difference distribution (see rows 487-491) can induce to reject the null hypothesis.

Concerning the XBT results, when normalized difference are calculated, taking into account the XBT uncertainty, the average positive bias of XBT measurements with respect to Argo ones has no practical consequence, when using XBT instruments instead of Argo, since it is well encompassed within the data standard deviation, i.e. the variability of the temperature differences (XBT-Argo).

*********************************************************

Request - Again, perhaps authors should present this more methodically in the Results section and explain the choices more carefully. For example, why are the authors dividing measurement uncertainty by 2?

Reply, rows 353 to 361 - " Therefore, the normalized differences were calculated between the two instruments, i.e. |t_XBT-t_Argo|/U(t_XBT), where U is the expanded uncertainty associated with the measurement of the whole XBT system, and checked how many were found to be lower than 1, hence indicating a satisfactory metrological agreement. For this purpose, neglecting the instrumental Argo uncertainty on temperature measurements (XBT temperature readings are intrinsically less accurate than Argo ones by a factor up to about 10 and similar conclusions for the depth sensors), a standard uncertainty of 0.1 °C was assigned to XBT measurements, obtained as a half of the overall XBT accuracy of 0.2 °C stated by the manufacturer [6,41]. This standard uncertainty can be considered as obtained “in field” (i.e. during working condition in sea, for a typical XBT launch from a traveling ship)."

*********************************************************

Request - I respectfully disagree with the conclusions presented herein: it looks to me that XBT (except T5) is consistently overestimating temperature, if we presume that Argo is closer to truth, ( see in Figure 10, Tables 5-7). What is
causing this consistent positive bias for XBT? Is this a depth calculation problem for XBT? Can it be resolved/altered with a depth calculation correction?

Reply - We do not deny the existence of an XBT warm bias in our results. We do not agree with the definition of "consistent positive bias" (at least below 100 m depth), being well within the XBT instrumental uncertainty. In our opinion it is not due to a depth calculation problem, but intrinsic in the XBT system.

See Ref. 41, section "Sippican Response to the Anderson Report".

See also Goes et al. The impact of improved thermistor calibration on the expendable bathythermograph priofile data, JOAT 2017, vol. 34 1947-1961. DOI: 10.1175/JETCH-D-17-0024.1

*********************************************************

Request - Here again, more explanation of the t-test results is needed. Let’s go back to Figure 14 – the histogram of pair-wise differences between Argo and CTD temperatures – it looks fairly “normal” or “gaussian” around a mean of
differences very close to 0 (left panel) and exactly 0 (right panel). This leads me to believe that the Argo vs CTD temperature comparisons are very good - with much of the remaining dispersion simply due to unconstrained time/space
mismatch – hence the histogram’s gaussian symmetry. Under the presumption of a normal probability density function, and the histogram looks normal, then in both cases (left and right) ~ 95% of the Argo v CTD temperature differences are within ~ +/- 0.4 degrees C. That strikes me as good agreement considering inherent imperfections in space/time match-ups within one Rossby radius of deformation.

Reply - Yes we agree with you. The very good agreement between Argo and CTD is mainly due to the selection criteria (i.e. only 1 day as time interval and QC flag equal to 1).

*********************************************************

Request - Yet, the t-test leads authors to reject the null hypothesis for nearly all of the Argo vs CTD comparisons (line 487). Are the authors concluding the Argo/CTD agreement is poor based strictly on pair-wise t-test p-values? Argo v CTD (t) appears much closer than XBT v Argo (t), and that appears at odds
with the mixed t-test results as they are presented in the discussion.

Reply - We gave the explanation in previous replies (see also sections (3.1.3. and 3.2.1.)

*********************************************************

Request - Are the mean differences in Figure 16 for each depth interval
larger or smaller than the delta (t) shown in the tables (5-7) for XBT v Argo ? And why not show histograms of XBT v Argo differences as well? Some uniformity/consistency of comparison graphs for the 2 sets of comparisons (XBT v Argo, and Argo v CTD) would help.

Reply - YES, (Argo-CTD) mean differences are smaller than (XBT-ARGO) differences. In order to homogenize the comparison diagram, we have substituted histograms in Fig. 14 and fig. 15 by corresponding box plot.

*********************************************************

Request - In summary – the t-tests and p-values stuffed into the Discussion section created confusion for me and appear at odds with more obvious conclusions that would follow from the earlier analysis presented in the results. If authors apply the same due diligence to the t-tests as they obviously have
with the remaining bulk of the analysis, I’m sure this can be easily corrected. If t-tests reach different conclusions for XBT v Argo than Argo v CTD, these conclusions need to be reconciled with the preceding analysis. The last section on Argo float stability should also be in the results section. Final paragraph/section of the paper should provide an overall summary of the important findings and their implications.

Reply - As already explained, t-test results have been moved to proper sections into Results paragraph.

*********************************************************

Additional Comments:

*********************************************************

Request - Strong linearity and very high linear correlation for measurement agreement (For example, Figures 18 and 19) are expected from ocean profile data (especially, temperature/density). For example, 2 temperature profiles could be apart by 10 degrees or more, and still have a nearly perfect linear correlation since the bulk profile variance is determined by depth. Authors might consider, in the future, a different metric of measurement agreement dispersion such as Root-Mean-Square-Difference.

Reply - In Rows 455-458 we added the following comment:"Due to the fact that both Argo and ship-based CTD host similar sensors, a very strong linearity and high linear correlation should be reasonably expected, even in cases when there is an actual bias between the profiles."

For what concerns the use of  a different metric, we agree: in the future it could be used for a work including also the uncertainty of pressure measurements, which is assumed to be without uncertainty in this paper.

*********************************************************

Request - Authors repeatedly make references to the ocean as a “thermostatic bath.” I do not follow this comparison. If deep ocean temperature does not change, then why are we measuring it?

Reply - In our paper, "thermostatic bath" conditions describe a quasi-static situation, occurring usually below a certain depth in a certain amount of time, for which the profiles acquired by two or more different instruments can be compared assuming the measurand (i.e. seawater temperature or salinity)  as constant.

Rows from 334 to 343 have been modified as follows: "Hence, it cannot be excluded that, in this case, the two instruments give the "same measure": this fact can be considered as a good indicator of the interchangeability of these two instruments, also indicating that, under these space-time conditions, the sea behaves reasonably like a thermostatic bath. This is enhanced by the seawater characteristics in the Mediterranean Sea (with a temperature range of about 1.0 °C even on 2-3 thousand meters of water) so that the temperature gradient is very small (frequently, some 10-3 °C m-1) making reasonable such an expression. In addition, when the dense water formation occurs in winter months (e.g. in Gulf of Lyon or South Adriatic Sea), from surface down to about 500 m depth (or more) XBTs, Argo floats and CTDs are all able to measure a variation for temperature values not greater than 0.02-0.03 °C and this makes that example self-explanatory."

*********************************************************

Request - Additional suggestion: upper ocean variance in temperature
may be due to diurnal surface warming during boreal spring/summer. Would a seasonal parsing of these data improve any < 100 m comparisons?

Reply, rows 560-564 - "On the other hand, in the 0-100 m depth region it can have even heavy consequences where the structures of the upper thermocline start, and gradients greater than 2 °C m-1 can be measured (while in the winter period the gradient is much smaller and XBTs are able to better describe the local water temperature, so that their bias and SD are homogeneous along the whole water column)."

*********************************************************

Request - Minor detail: lats/lons on Figure 2 are in very small font – might want to reduce tick interval and enlarge font.

Response-Figure 2 changed following Reviewer suggestions.

*********************************************************

Round 2

Reviewer 1 Report

< Specific comments>

This issue is a very increasingly important issue in oceanography or changes in the marine environment (such as water temperature or salinity). Especially, the seasonal change of ocean currents and Thermocline. In the manuscript, the data and methods are thoughtful and relatively well presented. But, there are several key problems the author still needs to address in the manuscript of R1. Now, the content of the manuscript has been more completely revised for the review recommendations of R1, including abstract, introduction, and research objectives and references in the manuscript.

Especially, the authors had proposed many specific suggestions provide according to comprehensively contents of international papers for the international audiences in the discussion section of the manuscript content.

I think this manuscript is ready for official acceptance and publication. However, I suggest that the author should conduct detailed final proofreading of the content of the manuscript before formal acceptance, including the information of authors, references, names of every section, English grammar, etc.